# SCHEDULING THE LEARNING RATE VIA HYPERGRA-DIENTS: NEW INSIGHTS AND A NEW ALGORITHM

## ABSTRACT

We study the problem of fitting task-specific learning rate schedules from the perspective of hyperparameter optimization. This allows us to explicitly search for schedules that achieve good generalization. We describe the structure of the gradient of a validation error w.r.t. the learning rate, the hypergradient, and based on this we introduce a novel online algorithm. Our method adaptively interpolates between the recently proposed techniques of Franceschi et al. (2017) and Baydin et al. (2018), featuring increased stability and faster convergence. We show empirically that the proposed method compares favorably with baselines and related methods in terms of final test accuracy.

## 1 INTRODUCTION

Learning rate (LR) adaptation for first-order optimization methods is one of the most widely studied aspects in optimization for learning methods — in particular neural networks — with early work dating back to the origins of connectionism (Jacobs, 1988; Vogl et al., 1988). More recent work focused on developing complex schedules that depend on a small number of hyperparameters (Loshchilov & Hutter, 2017; Orabona & Pál, 2016). Other papers in this area have focused on the optimization of the (regularized) training loss (Schaul et al., 2013; Baydin et al., 2018; Wu et al., 2018) . While quick optimization is desirable, the true goal of supervised learning is to minimize the generalization error, which is commonly estimated by holding out part of the available data for validation. Hyperparameter optimization (HPO), a related but distinct branch of the literature, specifically focuses on this aspect, with less emphasis on the goal of rapid convergence on a single task. Research in this direction is vast (see Hutter et al. (2019) for an overview) and includes model-based (Snoek et al., 2012; Hutter et al., 2015), model-free (Bergstra & Bengio, 2012; Hansen, 2016), and gradient-based (Domke, 2012; Maclaurin et al., 2015) approaches. Additionally, works in the area of learning to optimize (Andrychowicz et al., 2016; Wichrowska et al., 2017) have focused on the problem of tuning parameterized optimizers on whole classes of learning problems but require prior expensive optimization and are not designed to speed up training on a single specific task.

The goal of this paper is to automatically compute *in an online fashion* a learning rate schedule for stochastic optimization methods (such as SGD) only on the basis of the given learning task, aiming at producing models with associated small validation error. We study the problem of finding a LR schedule under the framework of gradient-based hyperparameter optimization (Franceschi et al., 2017): we consider as an optimal schedule $\eta^* = (\eta_0^*, \dots, \eta_{T-1}^*) \in \mathbb{R}_+^T$ a solution to the following constrained optimization problem

$$\min\{f_T(\eta) = E(w_T(\eta)) : \eta \in \mathbb{R}_+^T\} \quad \text{s.t.} \quad w_0 = \bar{w}, \quad w_{t+1}(\eta) = \Phi_t(w_t(\eta), \eta_t) \quad (1)$$

for $t = \{0, \dots, T-1\} = [T]$, where $E : \mathbb{R}^d \to \mathbb{R}_+$ is an objective function, $\Phi_t : \mathbb{R}^d \times \mathbb{R}_+ \to \mathbb{R}^d$ is a (possibly stochastic) weight update dynamics, $\bar{w} \in \mathbb{R}^d$ represents the initial model weights (parameters) and finally $w_t$ are the weights after $t$ iterations. We can think of $E$ as either the training or the validation loss of the model, while the dynamics $\Phi$ describe the update rule (such as SGD, SGD-Momentum, Adam etc.). For example in the case of SGD, $\Phi_t(w_t, \eta_t) = w_t - \eta_t \nabla L_t(w_t)$, with $L_t(w_t)$ the (possibly regularized) training loss on the $t$-th minibatch. The *horizon* $T$ should be large enough so that the training error can be effectively minimized, in order to avoid underfitting. Note that a too large value of $T$ does not necessarily harm since $\eta_k = 0$ for $k > \bar{T}$ is still a feasible solution, implementing early stopping in this setting.

Problem (1) can be in principle solved by any HPO technique. However, most HPO techniques, including those based on hypergradients Maclaurin et al. (2015) or on a bilevel programming formulation (Franceschi et al., 2018; MacKay et al., 2019) would not be suitable for the present purpose since they require multiple evaluations of $f$ (which, in turn, require executions of the weight optimization routine), thus defeating one of the main goals of determining LR schedules, i.e. speed. In fact, several other researchers (Almeida et al., 1999; Schraudolph, 1999; Schaul et al., 2013; Franceschi et al., 2017; Baydin et al., 2018; Wu et al., 2018) have investigated related solutions for deriving greedy update rules for the learning rate. A common characteristic of methods in this family is that the LR update rule does not take into account information from the future. At a high level, we argue that any method should attempt to produce updates that approximate the true and computationally unaffordable hypergradient of the *final* objective with respect to the current learning rate (in relation to this, Wu et al. (2018) discusses the bias deriving from greedy or short-horizon optimization). In practice, different methods resort to different approximations or explicitly consider greedily minimizing the performance after a single parameter update (Almeida et al., 1999; Schaul et al., 2013; Baydin et al., 2018). The type of approximation and the type of objective (i.e. the training or the validation loss) are in principle separate issues although comparative experiments with both objectives and the same approximation are never reported in the literature and validation loss is only used in the experiments reported in Franceschi et al. (2017).

One additional aspect needs to be taken into account: even when the (true or approximate) hypergradient is available, one still needs to introduce additional hyper-hyperparameters in the design of the online learning rate adaptation algorithm. For example in Baydin et al. (2018), hyper-hyperparameters include initial value of the learning rate $\eta_0$ and the hypergradient learning rate $\beta$. We find in our experiments that results can be quite sensitive to the choice of these constants.

In this work, we make a step forward in understanding the behavior of online gradient-based hyperparameter optimization techniques by (i) analyzing in Section 2 the structure of the true hypergradient that could be used to solve Problem (1) if wall-clock time was not a concern, and (ii) by studying in Section 3 some failure modes of previously proposed methods along with a detailed discussion of the type of approximations that these methods exploit. In Section 4, based on these considerations, we develop a new hypergradient-based algorithm which reliably produces competitive learning rate schedules aimed at lowering the final validation error. The algorithm, which we call MARTHE (Moving Average Real-Time Hyperparameter Estimation), has a moderate computational cost and can be interpreted as a generalization of the algorithms described in Baydin et al. (2018) and Franceschi et al. (2017). Unlike previous proposals, MARTHE is almost parameter-free in that it incorporates heuristics to automatically tune its configuration parameters (i.e. hyper-hyperparameters). In Section 5, we empirically compare the quality of different hypergradient approximations in a small scale task where true hypergradient can be exactly computed. In Section 6, we present a set of real world experiments showing the validity of our approach. We finally discuss potential future applications and research directions in Section 7.

## 2 STRUCTURE OF THE HYPERGRADIENT

We study the optimization problem (1) under the perspective of gradient-based hyperparameter optimization, where the learning rate schedule $\eta = (\eta_0, \dots, \eta_{T-1})$ is treated as a vector of hyperparameters and $T$ is a fixed horizon. Please refer to Appendix A for a summary of the main notation used throughout the paper. Since the learning rates are positive real-valued quantities, assuming both $E$ and $\Phi$ are smooth functions, we can compute the gradient of $f \in \mathbb{R}^T$, which is given by

$$\nabla f_T(\eta) = \dot{w}_T^\intercal \nabla E(w_T), \quad \text{where} \quad \dot{w}_T = \frac{\mathrm{d}w_T}{\mathrm{d}\eta} \in \mathbb{R}^{d \times T}, \tag{2}$$

where "$\intercal$" means transpose. The total derivative $\dot{w}_T$ can be computed iteratively with forward-mode algorithmic differentiation (Griewank & Walther, 2008; Franceschi et al., 2017) as

$$\dot{w}_0 = 0, \quad \dot{w}_{t+1} = A_t \dot{w}_t + B_t, \qquad \text{with} \qquad A_t = \frac{\partial \Phi_t(w_t, \eta_t)}{\partial w_t}, \quad B_t = \frac{\partial \Phi_t(w_t, \eta_t)}{\partial \eta}. \tag{3}$$

The Jacobian matrices $A_t$ and $B_t$ depend on $w_t$ and $\eta_t$, but we will leave these dependencies implicit to ease our notation. In the case of SGD[1], $A_t = I - \eta_t H_t(w_t)$ and $[B_t]_j = -\delta_{tj} \nabla L_t(w_t)^\intercal$, where

---

[1] Throughout we use SGD to simplify the discussion, however, similar arguments hold for any smooth optimization dynamics such as those including momentum terms.

subscripts denote columns (starting from 0), $\delta_{tj} = 1$ if $t = j$ and $0$ otherwise and $H_t$ is the Hessian of the training error on the $t-$th mini-batch [2]. We also note that, given the high dimensionality of $\eta$, reverse-mode differentiation would result in a more efficient (running-time) implementation. We use here forward-mode both because it is easier to interpret and visualize and also because it is closely related to the computational scheme behind MARTHE, as we will show in Section 4. Finally, we note that stochastic approximations of Eq. (2) may be obtained with randomized telescoping sums (Beatson & Adams, 2019) or hyper-networks based stochastic approximations (MacKay et al., 2019).

Eq. 3 describes the so-called tangent system (Griewank & Walther, 2008) which is a discrete affine time-variant dynamical system that measures how the parameters of the model would change for infinitesimal variations of the learning rate schedule, after $t$ iterations of the optimization dynamics. Notice that the "translation matrices" $B_t$ are very sparse, having, at any iteration, only one non-zero column. This means that $[\dot{w}_t]_j$ remains 0 for all $j \geq t$: $\eta_t$ affects only the future parameters trajectory. Finally, for a learning rate $\eta_t$, the derivative (a scalar) is

$$\frac{\partial f_T(\eta)}{\partial \eta_t} = [\nabla f_T(\eta)]_t = \left[ \left( \prod_{s=t+1}^{T-1} A_s \right) B_t \right]_t^\intercal \nabla E(w_T) = -\nabla L_t(w_t)^\intercal P_{t+1}^{T-1} \nabla E(w_T), \quad (4)$$

where the last equality holds true for SGD. Eq. (4) can be read as the scalar product between the gradients of the training error at the $t$-th step and the objective $E$ at the final iterate, *transformed by* the accumulated (transposed) Jacobians of the optimization dynamics, shorthanded by $P_{t+1}^{T-1}$. As it is apparent from Eq. (4), given $w_t$, the hypergradient of $\eta_t$ is affected only by the future trajectory and does not depend explicitly on $\eta_t$.

In its original form, where each learning rate is left free to take any permitted value, Eq. (1) represents a highly nonlinear setup. Although, in principle, it could be solved by projected gradient descent, in practice it is unfeasible even for small problems: evaluating the gradient with forward-mode is inefficient in time, since it requires maintaining a (large) matrix tangent system. Evaluating it with reverse-mode is inefficient in memory, since the entire weight trajectory $(w_i)_{i=0}^T$ should be stored. Furthermore, it can be expected that several updates of $\eta$ are necessary to reach convergence where each update requires computation of $f_T$ and the entire parameter trajectory in the weight space. Since this approach is computationally very expensive, we turn out attention to online updates where $\eta_t$ is required to be updated online based only on trajectory information up to time $t$.

## 3 ONLINE GRADIENT-BASED ADAPTIVE SCHEDULES

Before developing and motivating our proposed technique, we discuss two previous methods to compute the learning rate schedule online. The real-time hyperparameter optimization (RTHO) algorithm suggested in (Franceschi et al., 2017), reminiscent of stochastic meta-descent (Schraudolph, 1999), is based on forward-mode differentiation and uses information from the entire weight trajectory by accumulating partial hypergradients. Hypergradient descent (HD), proposed in (Baydin et al., 2018) and closely related to the earlier work of Almeida et al. (1999), aims at minimizing the loss w.r.t. the learning rate after one step of optimization dynamics. It uses information only from the past and current iterate.

Both methods implement update rules of the type

$$\eta_t = \max \left[ \eta_{t-1} - \beta \Delta \eta_t, 0 \right], \quad (5)$$

where $\Delta \eta_t$ is an online estimate of the hypergradient, $\beta > 0$ is a step-size or *hyper-learning rate* and the $\max$ ensures positivity[3]. To ease the discussion, we omit the stochastic (mini-batch) evaluation of the training error $L$ and possibly of the objective $E$.

---

[2]Note that techniques based on implicit differentiation (Pedregosa, 2016; Agarwal et al., 2017) or fixed-point equations (Griewank & Faure, 2002) (also known as recurrent backpropagation (Pineda, 1988)) cannot be readily applied to compute $\nabla f_T$ since the training loss $L$ does not depend explicitly on $\eta$.

[3]Updates could be also considered in the logarithmic space as done e.g. by Schraudolph (1999); we find it useful, however, to let $\eta$ reach 0 whenever needed, offering a natural way to implement early stopping in this context.

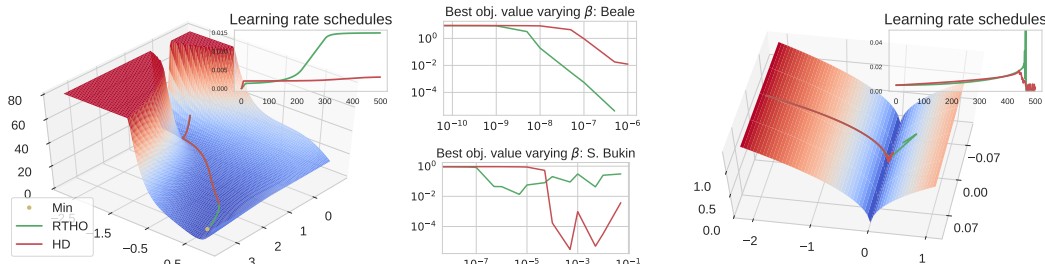

Figure 1: Loss surface and trajectories for 500 steps of gradient descent with HD and RTHO for Beale function (left) and (smoothed and simplified) Bukin N. 6 (right). Center: best objective value reached within 500 iterations for various values of $\beta$ that do not lead to divergence.

The update rules[4] are given by

$$\Delta^{\mathrm{RTHO}}\eta_t = \left[\sum_{i=0}^{t-1} P_{i+1}^{t-1} B_i\right]^{\intercal} \nabla E(w_t); \qquad \Delta^{\mathrm{HD}}\eta_t = B_{t-1}^{\intercal} \nabla E(w_t) \qquad (6)$$

for RTHO and HD respectively, where $P_t^{t-1} := I$. As it can be seen $\Delta^{\mathrm{RTHO}} = \Delta^{\mathrm{HD}} + r((w_i, \eta_i)_{i=0}^{t-2})$: the correction term $r$ can be interpreted as an "on-trajectory approximations" of longer horizon objectives as we will discuss in Section 4.

Although successful in some learning scenarios, we argue that both these update rules suffer from (different) pathological behaviors, as HD may be "shortsighted", being prone to underestimate the learning rate (as noted by Wu et al. (2018)), while RTHO may be too slow to adapt to sudden changes of the loss surface or, worse, may be unstable, with updates growing exponentially in magnitude. We exemplify these behaviors in Fig. 1, using two bidimensional test functions[5] from the optimization literature, where we set $E = L$ and we perform 500 steps of gradient descent, from a fixed initial point. The Beale function, on the left, presents sharp peaks and large plateaus. RTHO consistently outperforms HD for all probed values of $\beta$ that do not lead to divergence (Fig. 3 upper center). This can be easily explained by the fact that in flat regions gradients are small in magnitude, leading to $\Delta^{\mathrm{HD}}\eta_t$ to be small as well. RTHO, on the other hand, by accumulating all available partial hypergradients and exploiting second order information, is capable of making faster progress. We use a simplified and smoothed version of the Bukin function N. 6 to show the opposite scenario (Fig. 3 lower center and right). Once the optimization trajectory closes the valley of minimizers $y = 0.01x$, RTHO fails to discount outdated information, bringing the learning rate first to grow exponentially, and then to suddenly vanish to 0, as the gradient changes direction. HD, on the other hand, correctly damps $\eta$ and is able to maintain the trajectory close to the valley.

These considerations suggest that neither $\Delta^{\mathrm{RTHO}}$ nor $\Delta^{\mathrm{HD}}$ provide *globally useful* update directions, as large plateaus and sudden changes on the loss surface are common features of the optimization landscape of neural networks (Bengio et al., 1994; Glorot & Bengio, 2010). Our proposed algorithm smoothly and adaptively interpolates between these two methods, as we will present next.

## 4 OUR PROPOSAL: MARTHE

In this section, we develop and motivate MARTHE, an algorithm for computing LR schedule online during a single training run. This method maintains an adaptive moving-average over approximations of Eq. (4) of increasingly longer horizon, using the past trajectory and gradients to retain a low computational overhead. Further, we show that RTHO (Franceschi et al., 2017) and HD (Baydin et al., 2018) outlined above, can be interpreted as special cases of MARTHE, shedding further light on their behaviour and shortcomings.

---

[4] In (Franceschi et al., 2017) the authors the hyperparameter is updated every $K$ iterations. Here we focus on the case $K = 1$ which better allows for a unifying treatment. HD is developed using as objective the training loss $L$ rather than the validation loss $E$. We consider here without loss of generality the case of optimizing $E$.

[5] We use the Beale function defined as $L(x, y) = (1.5 - x + xy)^2 + (2.25 - x + xy^2)^2 + (2.625 - x + xy^3)^2$ and a simplified smoothed version of Buking N. 6: $L(x, y) = \sqrt{((y - 0.01x)^2 + \varepsilon)^{1/2} + \varepsilon}$, with $\varepsilon > 0$.

**Shorter horizon auxiliary objectives.** For $K > 0$, define $g_K(u, \xi)$, with $\xi \in \mathbb{R}_+^K$ as

$$g_K(u, \xi) = E(u_K(\xi)) \quad \text{s.t.} \quad u_0 = u, \quad u_{i+1} = \Phi(u_i, \xi_i) \text{ for } i = [K]. \tag{7}$$

The $g_K$s define a class of shorter horizon objective functions, indexed by $K$, which correspond to the evaluation of $E$ after $K$ steps of optimization, starting from $u \in \mathbb{R}^d$ and using $\xi$ as the LR schedule[6]. Now, the derivative of $g_K$ w.r.t. $\xi_0$, denoted $g_K'$, is given by

$$g_K'(u, \xi) = \frac{\partial g_K(u, \xi)}{\partial \xi_0} = [B_0]_0^\intercal P_1^{K-1} \nabla E(u_K) = -\nabla L(u)^\intercal P_1^{K-1} \nabla E(u_K), \tag{8}$$

where the last equality holds for SGD dynamics. Once computed on subsets of the original optimization dynamics $(w_i)_{i=0}^T$, the derivative reduces for $K = 1$ to $g_1'(w_t, \eta_t) = -\nabla E(w_{t+1}) \nabla L(w_t)^\intercal$ (for SGD dynamics), and for $K = T - t$ to $g_{T-t}'(w_t, (\eta_i)_{i=t}^{T-1}) = [\nabla f(\eta)]_t$. Intermediate values of $K$ yield cheaper, shorter horizon approximations of (4).

**Approximating the future trajectory with the past.** Explicitly using any of the approximations given by $g_K'(w_t, \eta)$ as $\Delta \eta_t$ is, however, still largely impractical, especially for $K \gg 1$. Indeed, it would be necessary to iterate the map $\Phi$ for $K$ steps in the future, with the resulting $(w_{t+i})_{i=1}^K$ iterations discarded after a single update of the learning rate. For $K \in [t]$, we may then consider evaluating $g_K'$ *exactly $K$ steps in the past*, that is evaluating $g_K'(w_{t-K}, (\eta_i)_{i=t-K}^{t-1})$. Selecting $K = 1$ is indeed equivalent to $\Delta^{\mathrm{HD}}$, which is computationally inexpensive. However, when past iterates are close to future ones (such as in the case of large plateaus), using larger $K$'s would allow in principle to capture longer horizon dependencies present in the hypergradient structure of Eq. 4. Unfortunately the computational efficiency of $K = 1$ does not generalize to $K > 1$, since setting $\Delta \eta_t = g_K'$ would require maintaining $K$ different tangent systems.

**Discounted accumulation of $g_k'$s.** The definition of the $g_K$s, however, allows one to highlight the recursive nature of the *accumulation* of $g_K'$. Indeed, by maintaining the vector tangent system

$$Z_0 = [B_0(u_0, \xi_0)]_0 \quad Z_{i+1} = \mu A_i(u_i, \xi_i) Z_i + [B_i(u_i, \xi_i)]_i \text{ for } i \geq 0, \quad Z_i \in \mathbb{R}^d, \tag{9}$$

computing $S_{K,\mu}(u, \xi) = \sum_{i=0}^{K-1} \mu^{K-1-i} g_{K-i}'(u_i, (\xi_j)_{j=i}^{K-1}) = Z_K^\intercal \nabla E(u_K)$ from $S_{K-1}$ requires only updating (9) and recomputing the gradient of $E$ for a total cost of $O(c(\Phi))$ per step both in time and memory using fast Jacobians vector products (Pearlmutter, 1994) where $c(\Phi)$ is the cost of computing the optimization dynamics (typically $c(\Phi) = O(d)$). The parameter $\mu \in [0, 1]$ allows to control how quickly past history is forgotten. One can notice that $\Delta^{\mathrm{RTHO}} \eta_t = S_{t,1}(w_0, (\eta_j)_{i=0}^{t-1})$, while $\mu = 0$ recovers $\Delta^{\mathrm{HD}} \eta_t$. Values of $\mu < 1$ help discounting outdated information, while as $\mu$ increases so does the horizon of the hypergradient approximations. The computational scheme of Eq. 9 is quite similar to that of forward-mode algorithmic differentiation for computing $\dot{w}$ (see Section 2 and Eq. 3); we note, however, that the "tangent system" in Eq. 9, exploiting the sparsity of the matrices $B_t$, only keeps track of the variations w.r.t the first component $\xi_0$, drastically reducing the running time.

**Adapting $\mu$ and $\beta$ online.** We may set $\Delta \eta_t = S_{t,\mu}(w_0, (\eta_j)_{i=0}^{t-1})$. This still would require choosing a fixed value of $\mu$, which should be validated on a separate set of held-out data. This may add an undesirable overhead on the optimization procedure. Furthermore, as discussed in Section 3, different regions of the loss surface may benefit from different effective approximation horizons. To address these issues, we propose to compute $\mu$ online. Ideally, we would like to verify that $\Delta \eta_t [\nabla f(\eta)]_t > 0$, i.e. whether the proposed update is a descent direction w.r.t. the true hypergradient. While this is unfeasible (since $\nabla f(\eta)$ is unavailable), we can cheaply compute, after the update $w_{t+1} = \Phi(w_t, \eta_t)$, the quantity

$$q(\mu_t) = \Delta \eta_{t+1} \cdot g_1'(w_t, \eta_t) = (\mu_t A_t Z_{t-1} + [B_t]_t)^\intercal \nabla E(w_{t+1}) \cdot [B_t]_t^\intercal E(w_{t+1}) \tag{10}$$

which, *ex post*, relates $\mu_t$ to the *one-step descent condition for $g_1$*. We set $\mu_{t+1} = h_\mu(q(\mu_t))$ where $h_\mu$ is a monotone scalar function with range in $[0, 1]$ (note that if $\mu_t = 0$ then Eq. 10 is non-negative). For space limitations, we defer the discussion of the choice of $h_\mu$ and the effect of adapting online the approximation horizons to the Appendix. We can finally define the update rule for MARTHE as

$$\Delta \eta_t = \sum_{i=0}^{t-1} \left( \prod_{j=i}^{t-2} \mu_j \right) g_{t-i}' \left( w_i, (\eta_j)_{j=i}^{t-1} \right). \tag{11}$$

---

[6] Note that, formally, $\xi$ and $u$ are different from $\eta$ and $w$ from the previous sections; later, however, we will evaluate the $g_K$'s on subsequences of the optimization trajectory.

We further propose to adapt $\beta$ online, implementing with this work a suggestion from Baydin et al. (2018). We regard the LR schedule as a function of $\beta$ and apply the same reasoning done for $\eta$, keeping $\mu = 0$, to avoid maintaining an additional tangent system which would involve third order derivatives of the training loss $L$. We then set $\beta_{t+1} = \beta_t - \beta \Delta \eta_{t+1} \cdot \Delta \eta_t$, where, with a little notation override, $\beta$ becomes a fixed step-size for adapting the hyper-learning rate. This may seem a useless trade-off at first; yet, as we observed experimentally, one major advantage of lifting the adaptive dynamics by one more level is that it injects additional stability in the learning system, in the sense that good values of this last parameter of MARTHE lays in a much broader range range than those of good hyper-learning rates. In fact, we observe that when $\beta$ is too high, the dynamics diverges within the first few optimization steps; whereas, when it does not, the final performances are rather stable.

Algorithm 1 presents the pseudocode of MARTHE. The runtime and memory requirements of the algorithm are dominated by the computation of the variables $Z$. Being these structurally identical to the tangent propagation of forward mode algorithmic differentiation, we conclude that the runtime complexity is up to four times that of the underlying optimization dynamics $\Phi$ and the memory requirement up to two times (see Griewank & Walther, 2008, Sec. 4). We suggest the default values of $\beta_0 = 0$ and $\eta_0 = 0$ when no prior knowledge of the task at hand is available. Finally, we suggest to wrap Algorithm 1 in a selection procedure for $\beta$, where one may start with a high enough $\beta$ and aggressively diminish it (e.g. decimating it) until the learning system does not diverge.

---

**Algorithm 1 MARTHE**; requires $\beta$, $\eta_0$, $\beta_0 = 0$

---
Initialization of $w, \eta$ and $Z_0 \leftarrow 0$, $\mu_0 \leftarrow 0$
**for** $t = 0$ **to** $T$ **do**
    $\eta_t \leftarrow \max \left[ \eta_{t-1} - \beta_t \Delta \eta_t, 0 \right]$                                {Update LR if $t > 0$}
    $Z_{t+1} \leftarrow \mu_t A_t(w_t, \eta_t) Z_t + [B_t(w_t, \eta_t)]_t$        {Tangent system update}
    $w_{t+1} \leftarrow \Phi_t(w_t, \eta_t)$                                          {Parameter update}
    $\mu_{t+1} \leftarrow h_\mu(q(\mu_t))$                              {Compute $\mu_t$, see Eq. 10}
    $\beta_{t+1} \leftarrow \beta_t - \beta \Delta \eta_{t+1} \cdot \Delta \eta_t$                       {Update hyper-LR}
**end for**

---

## 5 OPTIMIZED AND ONLINE SCHEDULES

In this section, we empirically compare the optimized LR schedules found by approximately solving Problem 1 by gradient descent (denoted LRS-OPT), where the hypergradient is given by Eq. 4, against HD, RTHO & MARTHE schedules where for MARTHE we consider both the adaptation schemes for $\mu$ and $\beta$ presented in the previous section as well as fixed hyper-learning rate and discount factor $\mu$. We are interested in understanding and visualizing the qualitative similarities among the schedules, as well as the effect of $\mu$ and $\beta$ and the adaptation strategies on the final performance measure. To this end, we trained feedforward neural networks, with three layers of 500 hidden units each, on a subset of 7000 MNIST (LeCun et al., 1998) images. We used the cross-entropy loss and SGD as the optimization dynamics $\Phi$, with a mini-batch size of 100. We further sampled 700 images to form the validation set and defined $E$ to be the validation loss after $T = 512$ optimization steps (about 7 epochs). For LRS-OPT, we randomly generated different mini-batches at each iteration to prevent the schedule from unnaturally adapting to a specific sample progression[7]. We initialized $\eta = 0.01 \cdot \mathbf{1}_{512}$ for LRS-OPT and set $\eta_0 = 0.01$ for all adaptive methods, and repeated the experiments for 20 random seeds (except LRS-OPT, repeated only for 4 seeds). Results are visualized in Figure 2.

Figure 2 (left) shows the LRS-OPT schedules found after 5000 iterations of gradient descent: the plot reveals a strong initialization (random seed) specific behavior of $\eta^*$ for approximately the first 100 steps. The LR schedule then stabilizes or slowly decreases up until around 50 iterations before the final time, at which point it quickly decreases (recall that with LRS-OPT all $\eta_i$, including $\eta_0$, are optimized "independently" and may take any permitted value). Figure 2 (center) present a qualitative comparison between the offline LRS-OPT schedule and the online ones. HD generates schedules that quickly decay to very small values, while RTHO schedule linger or fail to decrease, possibly causing instability and divergence in certain cases. Fixing $\mu = 0.99$ seems to produce schedules that

---

[7]We retained, however, the random initialization of the network weights, to account for the impact that this may have on the initial part of the trajectory (see Figure 2 (left)). This is done to offer a more fair comparison between LRS-OPT and online methods, which compute the trajectory only once.

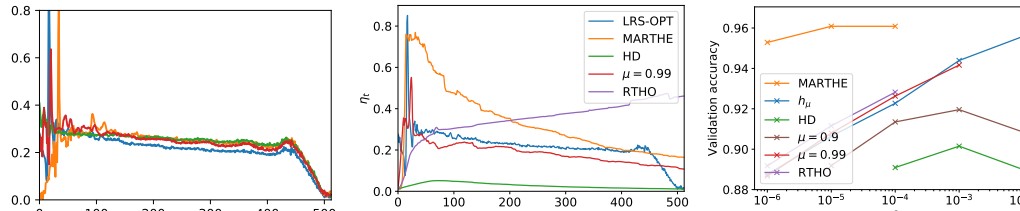

Figure 2: Left: schedules found by LRS-OPT (after 5000 iterations of SGD) on 4 different random seeds. Center: qualitative comparison between offline and online schedules for one random seed. For MARTHE with fixed $\mu$, we report the best performing one. For each method, we report the schedule generated with the value of $\beta$ that achieves the best average final validation accuracy. Plots for the remaining random seeds can be found in the appendix. Right: Average validation accuracy of over 20 random seeds, for various values of $\beta$. When no point is reported it means that the achieved average accuracy for that configuration falls below 88% (or diverged). For reference, the average validation accuracy of the network trained with $\eta = 0.01 \cdot \mathbf{1}_{512}$ is 87.5%, while LRS-OPT attains 96.2%.

remarkably mimic the the optimized one; yet, unfortunately, this happens only for a small range of values of $\mu$ which we expect to be task dependent. Using both the adaptation schemes for $\mu$ and $\beta$ (curve named MARTHE in the plots), allows to reliably find highly non-trivial schedules that capture the general behavior of the optimized one (additional plots in the Appendix).

Figure 2 (right) shows the average validation accuracy over 20 runs (rather than loss, for easier interpretation) of the online methods, varying $\beta$ and discarding values below 88% of validation accuracy. In particular, fixing $\mu > 0$ seems to have a beneficial impact for all tried values of the hyper-learning rate $\beta$. Using only the heuristic for adapting $\mu$ online (blue line, named $h_\mu$ in the plot) further helps, but is somewhat sensitive to the choice of $\beta$. Using both the adaptive mechanisms, beside improving the final validation accuracy, seems to drastically lower the sensitivity on the choice of this parameter, provided that the learning system does not diverge. Finally, we note that even with this very simple setup, a single run of LRS-OPT (which comprises 5000 optimization steps) takes more than 2 hours on an M-40 NVIDIA GPU. In contrast, all adaptive methods requires less than a minute to conclude (HD being even faster).

## 6 EXPERIMENTS

We run experiments with an extensive set of learning rate scheduling techniques. Specifically, we compare MARTHE against the following fixed LR scheduling strategies: (i) exponential decay (ED) – where the LR schedule is defined by $\eta_t = \eta_1 \gamma^t$ (ii) staircase decay and (iii) stochastic gradient descent with restarts (SGDR) by Loshchilov & Hutter (2017). Moreover, we compare against online strategies such as HD and RTHO. For all the experiments, we used a single Volta V100 GPU (AWS P3.2XL). We fix the batch-size at 128 samples for all the methods, and terminate the training procedure after a fixed number of epochs (200). We set $L$ as the cross entropy loss with weight-decay regularization (with factor of $5 \cdot 10^{-4}$) and set $E$ as the unregularized cross entropy loss on validation data. All the experiments with SGDM have an initial learning rate ($\eta_0$) of 0.1 and for Adam, we set it to $3 \cdot 10^{-4}$. For staircase, we decay the learning rate by 90% after every 60 epochs. For exponential decay, we fix a decay factor of 0.99 per epoch, and for SGDR we use $T_0 = 10$ and $T_{mult} = 2$. For HD and RTHO, we set the $\beta$ as $10^{-6}$ and $10^{-8}$ respectively. Momentum for SGDM is kept constant to 0.9. For Adam we used the standard values for the remaining configuration parameters. We run all the experiments with 5 different seeds reporting average and standard deviation, recording accuracy, loss value and generated learning rate schedules.

We trained image classification models on two different datasets commonly used to benchmark optimization algorithms for deep neural networks: CIFAR-10 (Krizhevsky et al., 2014) where we trained a VGG-11 (Simonyan & Zisserman, 2014) network with BatchNorm (Ioffe & Szegedy, 2015) using SGDM as the inner optimizer, and CIFAR-100 (Krizhevsky et al., 2014) where we trained a ResNet-18 (He et al., 2016) using Adam (Kingma & Ba, 2014). The source code in PyTorch and TensorFlow to reproduce the experiments will be made publicly available.

In Figures 3 and 4, we report the results of our experiments for CIFAR-10 with VGG-11 and CIFAR-100 with ResNet-18 respectively. For both figures, we report from left to right: accuracy in percentage, validation loss value, and an example of a generated learning rate schedule.

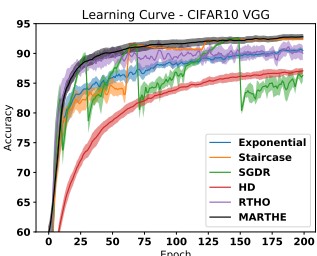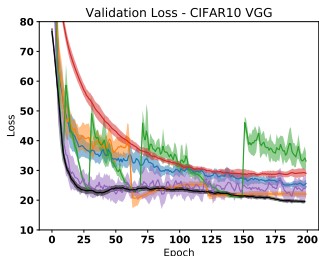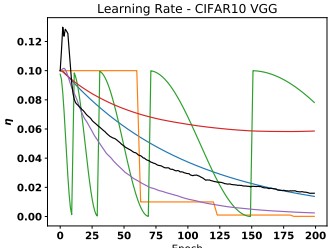

Figure 3: Results of VGG-11 on CIFAR-10, and SGDM as the inner optimizer concerning: (Left) accuracy, (Center) loss of the objective function on the validation set and (Right) generated learning rate schedule for each method.

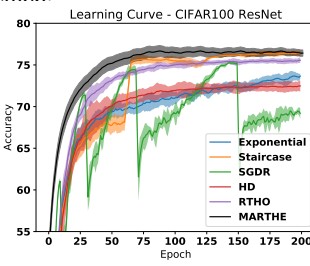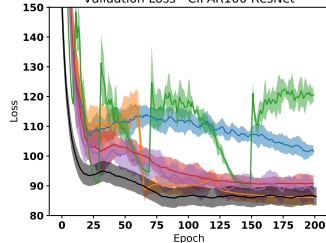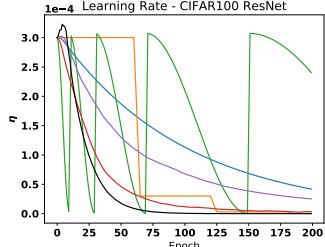

Figure 4: Results of ResNet-18 on CIFAR-100, and Adam as the inner optimizer concerning: (Left) accuracy, (Center) loss of the objective function on the validation set and (Right) generated learning rate schedule for each method.

MARTHE produces LR schedules that lead to trained models with very competitive final validation accuracy in both the experimental settings, virtually requiring no tuning. For setting the hyper-learning rate step-size of MARTHE we followed the simple procedure outlined at the end of Section 4, while for the other methods we performed a grid search to select the best value of the respective algorithmic parameters. On CIFAR-10, MARTHE obtains a best average accuracy of 92.79% statistically on par with SGDR (92.54%), while clearly outperforming the other two adaptive algorithms. On CIFAR-100, MARTHE leads to faster convergence during then whole training compared to all the other methods, reaching an accuracy of 76.68%, comparable to staircase schedule with 76.40%. We were not able to achieve competitive results with SGDR in this setting, despite trying several values of the two main configuration parameters within the suggested range. Further, MARTHE produces aggressive schedules (see Figure 3 and 4 right, for an example) that increase the LR at the beginning of the training, sharply decreasing it after a few epochs. We observe empirically that this leads to improved convergence speed and competitive final accuracy.

Additional experimental validation is reported in Appendix D and includes results for MARTHE with an initial learning rate $\eta_0 = 0$.

## 7  CONCLUSION

Finding a good learning rate schedule is an old but crucially important issue in machine learning. This paper makes a step forward, proposing an automatic method to obtain performing LR schedules that uses an adaptive moving average over increasingly long hypergradient approximations. MARTHE interpolates between HD and RTHO taking the best of the two worlds. The implementation of our algorithm is fairly simple within modern automatic differentiation and deep learning environments, adding only a moderate computational overhead over the underlying optimizer complexity.

In this work, we studied the case of optimizing the learning rate schedules for image classification tasks; we note, however, that MARTHE is a general technique for finding online hyperparameter schedules (albeit it scales linearly with the number of hyperparameters), possibly implementing a competitive alternative in other application scenarios, such as tuning regularization parameters (Luketina et al., 2016). We plan to further validate the method both in other learning domains for adapting the LR and also to automatically tune other crucial hyperparameters. We believe that another interesting future research direction could be to learn the adaptive rules for $\mu$ and $\beta$ in a meta learning fashion.

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

## A   TABLE OF NOTATION

Please refer to Table 1 for a summary and description of the notation used throughout the paper.

Table 1: Notation, with description and examples when appropriated, in order of appearance.

| Notation | Description | Examples and notes |
|---|---|---|
| $T \in \mathbb{N}$ | Total number of iterations, *horizon* | |
| $w \in \mathbb{R}^d$ | Model parameters, inner optimization variables | Weights of a neural network |
| $\eta = (\eta_0, \dots, \eta_{T-1})$ | Learning rate schedule (LRS), hyperparameter | |
| $f_T(\eta) \in \mathbb{R}^+$ | Response function after $T$ iterations (w.r.t. the LRS) | |
| $E(w) \in \mathbb{R}^+$ | Response function (w.r.t. the parameters) | Validation error |
| $L_t(w) \in \mathbb{R}^+$ | Training loss | |
| $\Phi_t(w_t, \eta_t) \in \mathbb{R}^d$ | Optimization (weight update) dynamics | SGD: $\Phi_t(w_t, \eta_t) = w_t - \eta_t \nabla L_t(w_t)$ |
| $\dot{w} \in \mathbb{R}^{d \times T}$ | Total derivative of $w$ w.r.t. $\eta$, variable of the tangent system | |
| $A_t = \frac{\partial \Phi_t(w_t, \eta_t)}{\partial w_t}$ | Jacobian of the dynamics w.r.t. the weights | SGD: $A_t = I - \eta_t H_t$ |
| $B_t = \frac{\partial \Phi_t(w_t, \eta_t)}{\partial \eta}$ | Jacobian of the dynamics w.r.t. the LRS | SGD: $[B_t]_j = -\delta_{tj} \nabla L_t(w_t)^\intercal$ |
| $P_r^s = \prod_{i=r}^s A_i$ | Product of Jacobians from iteration $s$ to $r$ | |
| $\Delta \eta_t$ | Update for the learning rate at iteration $t$ | |
| $\beta \in \mathbb{R}^+$ | Hyper-learning rate | |
| $\Delta^{\mathrm{HD}}$ | HD update | See (Baydin et al., 2018) |
| $\Delta^{\mathrm{RTHO}}$ | RTHO update | See (Franceschi et al., 2017) |
| $g_k(u, \xi) \in \mathbb{R}^+$ | Shorter horizon objectives, with horizon $K$, $u$ starting point and $\xi \geq 0 \in \mathbb{R}^K$ LRS | |
| $g'(u, \xi)$ | Derivative of $g_K$ w.r.t. $\xi_1$ | |
| $\mu$ | Dampring factor | |
| $Z \in \mathbb{R}^d$ | Variables of the vector tangent system | Cf. tangent matrix $\dot{w}$ |
| $S_{k,\mu}(u, \eta) \in \mathbb{R}$ | Online accumulation of shorter horizon derivatives $g'_K$ | |
| $h_\mu$ | Heuristics for dampening factor $\mu$ | See Appendix B |

## B   CHOICE OF HEURISTIC FOR ADAPTING $\mu$

We introduced in Section 4 a method to compute online the dampening factor $\mu_t$ based on the quantity

$$q(\mu_t) = \Delta \eta_{t+1} \cdot g'_1(w_t, \eta_t) = (\mu_t A_t Z_{t-1} + [B_t]_t)^\intercal \nabla E(w_{t+1}) \cdot [B_t]_t^\intercal E(w_{t+1}).$$

We recall that if $q(\mu_t)$ is positive then the update $\Delta \eta_{t+1}$ is a descent direction for the one step approximation of the objective $f_T$. We describe here the precise heuristic rule that we use in our experiments. Call $\tilde{q}(\mu_t) = \max(\min(q(\mu_t) g'_1(w_t, \eta_t)^{-2}, 1), 0) \in [0, 1]$ the normalized, thresholded $q(\mu_t)$. We propose to set

$$\mu_{t+1} = h_\mu(\mu_t) = \tilde{q}(\mu_t)^{\frac{1}{c_t+1}} \quad \text{with} \quad c_0 = 0, \quad c_{t+1} = \mu_t(1 + c_t),$$

where $c_t$ acts as a multiplicative counter, measuring the effective approximation horizon. The resulting heuristics is independent on the initialization of $\mu$ since $Z_0 = 0$. We note that whenever $\mu_t$ is set to 0, the previous hypergradient history is forgotten. Applying this heuristic to the optimization of the two test functions of Sec. 3 reveals to be successful: for the Beale function, $h_\mu$ selects $\mu_t = 1$ for all $t$, while for the smoothed Bukin, it selects $\mu_t = 0$ for around $40\%$ of the iterations, bringing down the minimum optimality gap at $10^{-6}$ for $\beta = 0.0005$.

We conducted exploratory experiments with variants of $h_\mu$ which include thresholding between -1 and 1 and "penalizing" updates larger than $g_1'$ without observing statistically significant differences. We also verified that randomly setting $\mu_t$ to 0 does not implement a successful heuristics, while introducing another undesirable configuration parameter. We believe, however that there is further space of improvement for $h_\mu$ (and possibly to adapt the hyper-learning rate), since $g_1'$ does not necessarily capture the long term dependencies of Problem 1. Meta-learning these update rules could be an interesting direction that we leave to future investigation.

## C  Optimized and Online Schedules: Additional Details

We show in Figure 5 the LR schedules for the experiments described in Section 5 for the remaining random seeds. The random seed controls the different initial points $w_0$, which is the same for all online methods and for LRS-OPT, and determines the mini-batch progression for the online methods (while for LRS-OPT the mini-batch progression is randomized at each outer iteration).

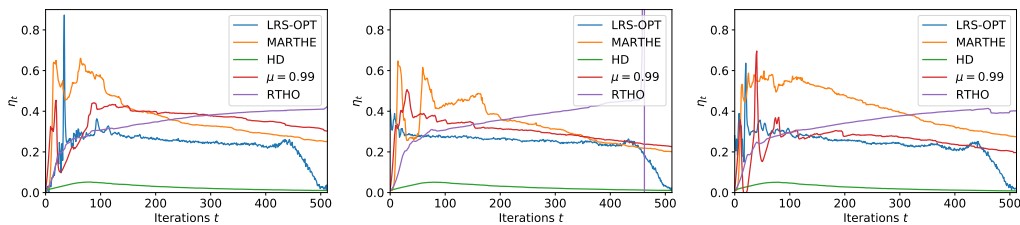

Figure 5: Comparison between optimized and online schedules for the remaining three seeds. For each method, we report the schedule generated with the hyper-learning rate (or step-size for adapting it) that achieves the best final validation accuracy.

## D  Additional Experimental Results

We report in this section additional experimental results to complement the analysis of Section 6.

Figure 6 shows accuracy, validation loss and learning rates schedules for CIFAR-100 dataset with a ResNet-18 model and SGDM as (inner) optimization methods. In accordance with previous experimental validation (Zagoruyko & Komodakis, 2016; Huang et al., 2017) we set the initial learning rate $\eta_0$ to 0.1 for all methods. We include also relevant statistics for MARTHE$_0$, that is MARTHE obtained by letting the schedule start at 0 (cyan lines in the plots). Setting $\eta_0 = 0$ is clearly not an optimal choice, nevertheless MARTHE is able to obtain competitive results also in this disadvantaged scenario, producing schedules that quickly reach high values and then sharply decrease within the first 40 epochs. Figure 6 (left) reports a sample of generated schedule. It is important to highlight how the heuristic methods, such as exponential decay, are not able to handle $\eta_0 = 0$. In fact, for non-adaptive methods, $\eta_0$ is indeed another configuration parameter that must be tuned. MARTHE$_0$, on the other hand, constitute a virtually parameterless method ($\beta$ can be quickly found with the strategy outlined at the end of Section 4) that can be employed in situations where we have no prior knowledge of the task at hand. Conversely as noted by Maclaurin et al. (2015) and Metz et al. (2019), too high (initial) learning rates are not well suited for gradient-based adaptive strategies: instability of the inner optimization dynamics indeed propagates to the hypergradient computation, possibly leading to "exploding" hypergradients.

Finally, we tried another configuration of parameters for SGDR, in order to make the last restart completing the training exactly after 200 epochs. We selected $T_0 = 10$ and $T_{mul} = 2.264$ (i.e. restarts at $10, 33, 84, 200$). Figure 7 and Figure 8 report the same experiments of Section 6 with this slightly different baseline. We note that the performance of SGDR remains similar.

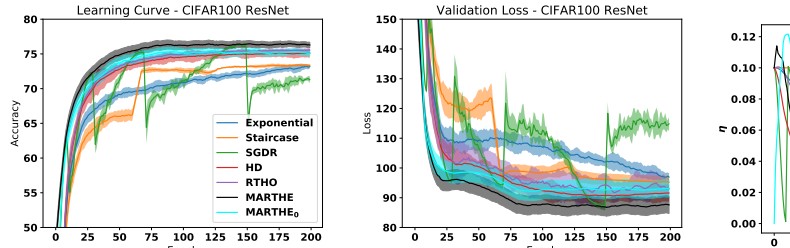

Figure 6: Results of ResNet-18 on CIFAR-100, and SGDM as the inner optimizer concerning: (Left) accuracy, (Center) loss of the objective function on the validation set and (Right) generated learning rate schedule for each method. In cyan we report the results for MARTHE with $\eta_0 = 0$.

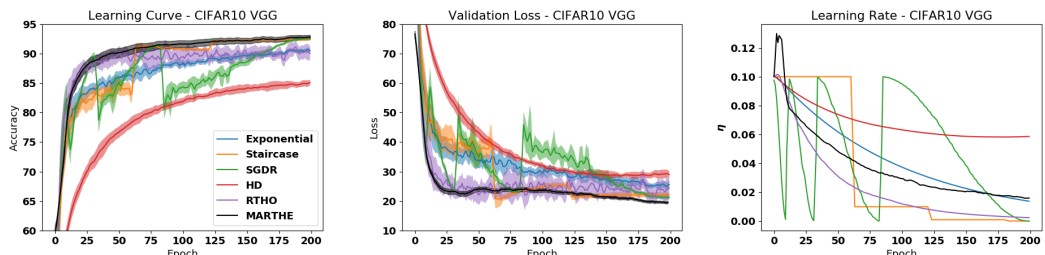

Figure 7: Results of VGG-11 on CIFAR-10, and SGDM as the inner optimizer concerning: (Left) accuracy, (Center) loss of the objective function on the validation set and (Right) generated learning rate schedule for each method.

# E SENSITIVITY ANALYSIS OF INADAPTIVE MARTHE WITH RESPECT TO $\eta_0$, $\mu$ AND $\beta$

In this section, we study the impact of $\eta_0$, $\mu$ and $\beta$ for MARTHE, when our proposed online adaptive methodologies for $\mu$ and $\beta$ are not applied. We think that the sensitivity of the methods is very important for the HPO algorithms to work well in practice, especially when they depend on the choice of some (new) hyperparameters such as $\mu$ and $\beta$.

We show the sensitivity of inadaptive MARTHE with respect to $\eta_0$ and $\mu$, fixing $\beta$. We used VGG-11 on CIFAR-10 with SGDM as optimizer, but similar results can be obtained in the other cases. Figure 9 shows the obtained test accuracy fixing $\beta$ to $10^{-7}$ (Left) and $10^{-8}$ (Right). The plots show a certain degree of sensitivity, especially with respect to the choice of a (fixed) $\mu$. This suggest that implementing adaptive strategies to compute online the dampening factor $\mu$ and the hyper-learning rate $\beta$ constitute an essential factor to achieve the competitive results reported in Section 6 and Appendix D.

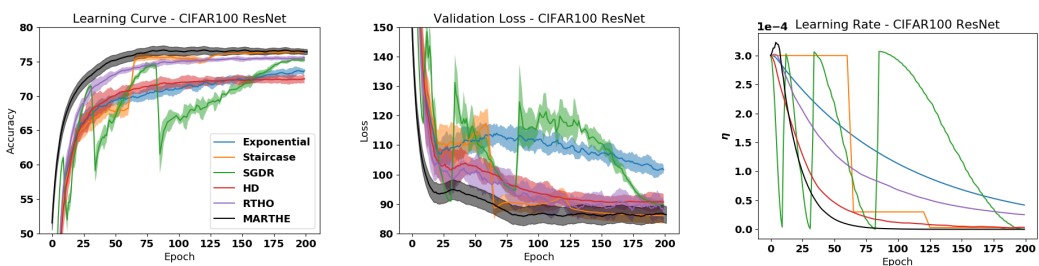

Figure 8: Results of ResNet-18 on CIFAR-100, and Adam as the inner optimizer concerning: (Left) accuracy, (Center) loss of the objective function on the validation set and (Right) generated learning rate schedule for each method.

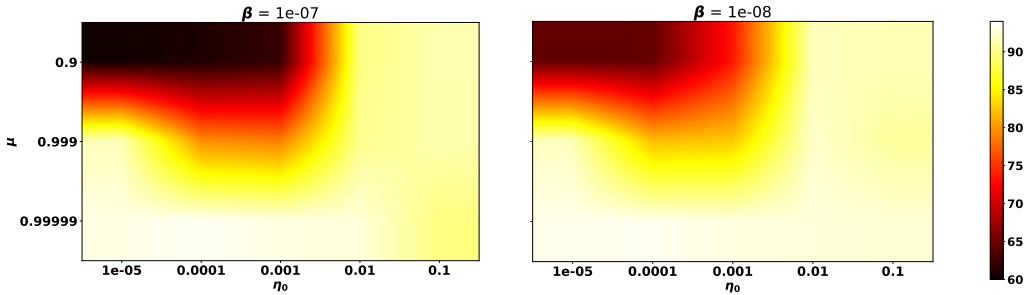

Figure 9: Sensitivity analysis of inadaptive MARTHE with respect to $\eta_0$ and $\mu$ fixing the value of $\beta$ to $10^{-7}$ (Left) and $10^{-8}$ (Right). We used VGG-11 on CIFAR-10 with SGDM as optimizer. Darker colors mean lower final accuracy.

