# OpenReview forum: "Scheduling the Learning Rate Via Hypergradients: New Insights and a New Algorithm"
_ICLR.cc/2020/Conference — Reject_

### Official Review · AnonReviewer1 · 2019-10-23
**Official Blind Review #1**

**Rating:** 1

**Review:**

The results are given only for the CIFAR datasets. Even for these two datasets the authors use very outdated networks, e.g., the best error rate for CIFAR-10 is in order of 6 percent. One should use contemporary/bigger networks, e.g., WRNs published in 2016 would give you about 4 percent.
1) The initial learning rate for CIFAR-10 and CIFAR-100 are different, respectively 0.1 and 0.0003. The use of such small initial learning rate for CIFAR-100 is not motivated especially given that it is usually in order of 0.05 or 0.1 when resnets are considered.
2) The authors don't compare to cosine annealing without restarts which is a pretty strong baseline.
3) The authors compare to SGDR but don't set its initial number of epochs in a way that its last restart convergences at around 200 epochs.
4) The proposed method has its own hyperparameters which greatly influence the results as shown in the appendix. I suspect that setting these hyperparameters is exactly what controls the slope of the learning schedule.

Overall, the results are not convincing. The authors show that the previous adaptive approaches don't work well on the CIFAR datasets (despite the fact that their authors claimed the oppositve) and I don't think that the paper contains enough material to avoid the situation that futures approaches will claim similar things about the current study.

**Experience Assessment:**

I have published in this field for several years.

**Review Assessment: Checking Correctness Of Derivations And Theory:**

I assessed the sensibility of the derivations and theory.

**Review Assessment: Checking Correctness Of Experiments:**

I assessed the sensibility of the experiments.

**Review Assessment: Thoroughness In Paper Reading:**

I read the paper at least twice and used my best judgement in assessing the paper.

---

> ### Author Response · Authors · 2019-11-08
> **Answer to Reviewer 1**
>
> In the following we try to address the reviewer’s concerns point by point.
> 1) The learning rates are indeed different because the (inner) optimization methods are different: SGDM on CIFAR10 and Adam on CIFAR100; please see 2nd and 3rd paragraph of Section 6. As it is well known SGDM and Adam have different ranges. Our aim was to showcase the behaviour of MARTHE with different optimization methods. For the sake of completeness, we added in the supplementary material of the updated version of the paper the results of the same experiments using SGDM for CIFAR100, and applying the suggested initial learning rate of 0.1 (see Section C).
> 2) We compared with SGDR that yields systematically better results than cosine annealing [1] at the same computational cost.
> 3) For CIFAR10 we used the best found hyperparameters by the authors [1]. We however tried to repeat the SGDR experiments obtaining the last convergence at epoch 200 which didn’t result in any statistical improvement in the best accuracy reached. For example, in the case of CIFAR10, our reported result has a best accuracy for SGDR of 92.54% vs. 92.36% using t0=10 and t_mul=2.264 (which reach the last convergence at epoch 200).
> 4) Please note that MARTHE has only one effective configuration parameter (beta) which is the step-size to adapt the hyper-learning rate. As we show in Section 5 and also empirically in Section 6, this parameter is quite easy to set: the method diverges very quickly for higher values of beta and when it starts converging it has consistent and stable results (see second last paragraph of Section 5). In fact, we propose at the end of Section 4 a very simple methodology to set this configuration parameter which we use in the experiments in Sec. 6.
>
> Regarding the results with previous adaptive approaches, to the best of our knowledge we are not aware of experiments with RTHO on CIFAR datasets. The results reported in [2], instead, are obtained with a slightly different architecture (VGG16 while we use a VGG11) and the statistics reported in [2] are different (they report validation loss while we report accuracy).
>
> As a final remark let us stress upon the fact that the main contribution of this paper is to better understand existing methods (HD and RTHO), to show that they have limitations and can fail in some cases (Sec 3 & Figure 1) and to demonstrate that they can be generalized using a single algorithm (Sec 4) which can interpolate between them to mitigate some of their limitations. The goal is not to win yet another performance battle but rather to improve the understanding on the topic, which is an important one in the context of training deep neural networks.
>
> [1] Loshchilov, Ilya, and Frank Hutter. "Sgdr: Stochastic gradient descent with warm restarts." arXiv preprint arXiv:1608.03983 (2016).
> [2] Baydin, Atilim Gunes, Robert Cornish, David Martinez Rubio, Mark Schmidt, and Frank Wood. "Online learning rate adaptation with hypergradient descent." arXiv preprint arXiv:1703.04782 (2017).

---

> > ### Comment · AnonReviewer1 · 2019-11-14
> > **Re**
> >
> > >> and applying the suggested initial learning rate of 0.1 (see Section C).
> >
> > It is 0.05 for all experiments with SGDR in [1]. The origin of 3e-4 for Adam is not clear to me.
> >
> > >> 2) We compared with SGDR that yields systematically better results than cosine annealing [1] at the same computational cost.
> >
> > They yield comparable results, see Table 1 in [1].
> >
> > >>  For example, in the case of CIFAR10, our reported result has a best accuracy for SGDR of 92.54% vs. 92.36% using t0=10 and t_mul=2.264 (which reach the last convergence at epoch 200).
> >
> > Why you didn't update Figure 1 with comparable SGDR and cosine annealing?
> >
> > >> t0=10 and t_mul=2.264
> > you could use t0=13 and t_mul=2 to have restarts after 13, 39, 91 and 195 epochs to avoid rounding issues.

---

> > > ### Author Response · Authors · 2019-11-14
> > > **Comments**
> > >
> > > Thank you for your comments. We reply below:
> > >
> > > >> It is 0.05 for all experiments with SGDR in [1]. The origin of 3e-4 for Adam is not clear to me.
> > > As you stated in your previous answer, the commonly used initial learning rate values for SGDM with resnet on CIFAR are both 0.1 and 0.05. We picked 0.1, and we run all the experiments and comparisons keeping this value fixed.
> > > Concerning Adam, the commonly used range is between 10-3 and 10-4. It is well known that Adam does not perform well with an initial learning rate of 0.1.
> > >
> > > >> They yield comparable results, see Table 1 in [1].
> > > According to the results in [1], cosine annealing underperforms on CIFAR100 (compared to SGDR), and it yields equivalent performance on CIFAR10. To keep the number of experiments at a reasonable number we decided to use SGDR as it is a newer LR scheduler compared to cosine annealing.
> > >
> > > >> Why you didn't update Figure 1 with comparable SGDR and cosine annealing?
> > > We do not understand this comment. Figure 1 shows the pitfalls of HD and RTHO on two synthetic test functions from the optimization literature. We remark that the aim of that section is to highlight the behavior of two previously proposed gradient-based algorithms which MARTHE generalizes.
> > >
> > > >> you could use t0=13 and t_mul=2 to have restarts after 13, 39, 91 and 195 epochs to avoid rounding issues.
> > > Following your earlier suggestion, we chose a schedule to terminate the last restart exactly at the last epoch (after rounding our restarts are at 10, 33, 84, 200). For sure, there are several possible choices for t0 and t_mul to obtain this behavior of the learning rate schedule. We would like to remark that this is indeed one of the advantages of using MARTHE, that is, it does not require the calibration of multiple configuration parameters.

---

> > > > ### Comment · AnonReviewer1 · 2019-11-14
> > > > **Re**
> > > >
> > > > >> We do not understand this comment. Figure 1
> > > > Sorry, it was a typo. I referred to Figure 4.
> > > >
> > > >  >> It is well known that Adam does not perform well with an initial learning rate of 0.1.
> > > > Yes, in my original comment I didn't notice that the results of CIFAR-100 were for Adam.
> > > >
> > > > >> To keep the number of experiments at a reasonable number we decided to use SGDR as it is a newer LR scheduler compared to cosine annealing.
> > > >
> > > > AFAIK, cosine annealing was introduced in that same paper. In fact, it is a parameter choice - when you don't perform restarts you end up with cosine annealing.
> > > >
> > > > >> We would like to remark that this is indeed one of the advantages of using MARTHE, that is, it does not require the calibration of multiple configuration parameters.
> > > >
> > > > It requires learning rate and MARTHE's hyperparameter to be calibrated in contrast to calibrating learning rate of cosine annealing.

---

### Official Review · AnonReviewer2 · 2019-10-26
**Official Blind Review #2**

**Rating:** 6

**Review:**

In this paper, the authors introduce a hypergradient optimization algorithm for finding learning rate schedules that maximize test set accuracy. The proposed algorithm adaptively interpolates between two recently proposed hyperparameter optimization algorithms and performs comparably in terms of convergence and generalization with these baselines.

Overall the paper is interesting, although I found it a bit dense and hard to read. I frequently found myself having to scroll to different parts of the paper to remind myself of the notation used and the definition of the different matrices. This makes it harder to evaluate the paper properly. The proposed algorithm seems interesting however, and the experimental results look quite impressive.

I have a few concerns regarding the experiments however, which explains my score:

1. In figure 2, does MARTHE diverge for values of beta greater than 1e-4? This seems to indicate that MARTHE is somehow more sensitive to beta than the other variations used. Do the authors have any intuition about what might be causing this behavior?

2. The initial learning rate for SGDM and Adam was fixed at certain values for all experiments. Why is this a reasonable thing to do? It feels like MARTHE should be compared to SGDM and Adam at least when the initial learning rate is tuned for these properly. Otherwise, it doesn't feel like a fair evaluation? To the best of my knowledge, the final achieved accuracies achieved with MARTHE however seem quite competitive with the best results typically reached with tuned SGDM on the convolutional nets used in the paper.

3. The learning rate schedules found by MARTHE seem to be somewhat counterintuitive. While an initial increase matches the heuristic of warmup learning rates frequently used when training convnets, the algorithms seems to decrease down the learning rate after that even quicker than what the greedy algorithm HD does. Do the authors have any intuition why this can lead to such a big improvement in performance over HD?

4. Is it possible to provide some sort of estimate of how much computation MARTHE requires compared to a single SGDM run? How feasible is to test this algorithm on a bigger classification model on ImageNet?

I think this paper is borderline, although I am leaning towards accepting it given the impressive empirical results. It would really improve the paper if the readability was improved, as well as if larger experimental results were included.

====================================

Edit after rebuttal:
I thank the authors for their response. I am happy with their response and am sticking to my score.

**Experience Assessment:**

I have read many papers in this area.

**Review Assessment: Checking Correctness Of Derivations And Theory:**

I assessed the sensibility of the derivations and theory.

**Review Assessment: Checking Correctness Of Experiments:**

I carefully checked the experiments.

**Review Assessment: Thoroughness In Paper Reading:**

I made a quick assessment of this paper.

---

> ### Author Response · Authors · 2019-11-08
> **Answer to Reviewer 2**
>
> We thank the reviewer for the helpful feedback. We answer below to the concerns raised.
>
> 1) Yes, you are correct. In Figure 2 (right) when no point is reported it means that the achieved average accuracy for that configuration falls below 88% or that the method diverged. We changed the caption to make this point clearer. Please note that the sensitivity of beta is different among the different methods since for MARTHE it controls the hyper-learning rate updates while for the others (HD, RTHO) is the hyper-learning rate itself. We have found empirically that the parameter beta for MARTHE is quite easy to set since the method either diverges very quickly for higher than appropriate values of beta or — when it starts converging —  it has consistent and stable results. See Fig. 2 (right) and Sec. 5 and 6. In fact, we propose at the end of Sec. 4 a very simple methodology to set this configuration parameter which we use in the experiments in Sec. 6.
> 2) In general, we agree with your comment. In this case, however, we decided to use the abundance of previous successful experimental results on the CIFAR10 and 100 datasets and thus used established settings from literature, e.g. [1,2,3].
> 3) We think that the initial increase of the LR brings the weights to a good initial point where, even with a quick exponential decrease, the method leads to very good results. This is somewhat in line with the intuition behind the super convergence effect on neural networks [4].
> 4) Yes. The computation of the variables Z is structurally identical to the tangent propagation of forward mode algorithmic differentiation. This means that theoretically the runtime complexity is up to 4 times that of the underlying optimization iteration and the memory requirement up to 2 times. We added a comment on this at the end of section 4. However, with our implementation (in PyTorch), we have noticed the running time to be roughly 5X slower for VGG and also observed that the slowness increases with the depth of the network (e.g. it is ~8X slower for ResNet). Due to this computational bottleneck, we have not yet been able to train on ImageNet dataset within a reasonable time-limit.
>
> Regarding the clarity of the paper, we would be very happy to improve the readability of our work in the final version. We kindly ask the reviewer to point us which sections/paragraphs or choices of notation need to be revised.
>
> [1] Huang, G., Liu, Z., Van Der Maaten, L., & Weinberger, K. Q. (2017). Densely connected convolutional networks. In Proceedings of the IEEE conference on computer vision and pattern recognition (pp. 4700-4708).
> [2] Zagoruyko, S., & Komodakis, N. (2016). Wide residual networks. arXiv preprint arXiv:1605.07146.
> [3] Loshchilov, I., & Hutter, F. (2018). Decoupled weight decay regularization.
> [4] Smith, Leslie N., and Nicholay Topin. "Super-Convergence: Very Fast Training of Neural Networks Using Large Learning Rates." arXiv preprint arXiv:1708.07120 (2017).

---

### Author Response · Authors · 2019-11-15
**Uploaded new revision**

Dear all,

we uploaded a revised version of the paper which includes:

1) a new experiment in Appendix D that shows the performances of MARTHE on CIFAR-100 with the initial learning rate set to 0 ($\eta_0 = 0$). MARTHE produces a competitive schedule also in this disadvantaged setting, empirically showing that the method is not particularly sensitive also to the initial learning rate (we remind that we provide a rule for computing online the damping factor $\mu$ and we find $\beta$ quickly as the first value that does not lead to divergence).
2) runs of SGDR with the last restart at epoch 200  in Figures 7 and 8, as asked by R1.
3) a table of notation with descriptions and examples when appropriated, in Appendix A. We hope this will improve the readability of the paper, answering some concerns expressed by R2.
4) a modified description of experiments in Appendix E to better underline that they refer to the `inadaptive` version of MARTHE.
5) a comment on the runtime and space complexity of MARTHE (as requested by R2).
6) links to the appendix in the main text and corrections of minor typos.

We hope that these additions and modifications will strengthen the submission and address the reviewers concerns.

We believe that this work makes a step forward (yet, certainly not the last) in the old but crucial topic of finding good learning rates, providing novel insights and a unified view of two previously proposed gradient-based adaptive algorithms. We developed a new method with the purpose of taking ``the best of both world'', providing a mathematical exposition of its internal functioning. In the experimental validation, we offered comparisons with two previous algorithms and highly performing non-adaptive baselines, arose in multiple years of experience with vision datasets such as CIFAR-10 and 100.

We thank the reviewers and the area chair for their time and valuable suggestions,

Sincerely,
The authors

---

### Decision · Program_Chairs · 2019-12-19

**Decision:**

Reject

**Comment:**

First, I'd like to apologize once again for failing to secure a third reviewer for this paper. To compensate, I checked the paper more thoroughly than standard.

The area of online adaptation of the learning rate is of great importance and I appreciate the authors' effort in that direction. The authors carefully abundantly cite the research on gradient-based hyperparameter optimization but I would have appreciated to also see past works on stochastic line search (for instance  "A stochastic line-search method with convergence rate") or statistical methods ("Using Statistics to Automate Stochastic Optimization").

The issue with these methods is that, despite usually very positive claims in the paper, they are not that competitive against a carefully tuned fixed schedule and end up not being used in practice. Hence, it is critical to develop a convincing experimental section to assuage doubts. Unfortunately, the experimental section of this work is a bit lacking, as pointed by both reviewers. I would like to comment on two points specifically:
- First, no plot uses wall-clock time as the x-axis. Since the authors state that it can be up to 4 times as slow per iteration, the gains compared to a carefully tuned schedule are unclear.
- Second, the use of a single (albeit two variants) dataset also leads to skepticism. Datasets have vastly different optimization properties and, by not using a wide range of them, one can miss the true sensitivity of the proposed algorithm.

While I do not think that the paper is ready for publication, I feel like there is a clear path to an improved version that could be submitted to a later conference.